



# Single precision arithmetic in ECHAM radiation reduces runtime and energy consumption

Alessandro Cotronei[1] and Thomas Slawig[1]

[1]Kiel Marine Science (KMS) – Centre for Interdisciplinary Marine Science, Dep. of Computer Science, Kiel University, 24098 Kiel, Germany.

**Correspondence:** Thomas Slawig, ts@informatik.uni-kiel.de

**Abstract.** We converted the radiation part of the atmospheric model ECHAM to single precision arithmetic. We analyzed different conversion strategies and finally used a step by step change of all modules, subroutines and functions. We found out that a small code portion still requires higher precision arithmetic. We generated code that can be easily changed from double to single precision and vice versa, basically using a simple switch in one module. We compared the output of the single precision version in the coarse resolution with observational data and with the original double precision code. The results of both versions are comparable. We extensively tested different parallelization options with respect to the possible performance gain, in both coarse and low resolution. The single precision radiation itself was accelerated by about 40%, whereas the speed-up for the whole ECHAM model using the converted radiation achieved 18% in the best configuration. We further measured the energy consumption, which could also be reduced.

## 1 Introduction

The atmospheric model ECHAM was developed at the Max Planck Institute for Meteorology (MPI-M) in Hamburg. Its development started in 1987 as a branch of a global weather forecast model of the European Centre for Medium-Range Weather Forecasts (ECMWF), thus leading to the acronym (EC for ECMWF, HAM for Hamburg). The model is used in different Earth System Models (ESMs) as atmospheric component, e.g., in the MPI-ESM also developed at the MPI-M, see Figure 1. The current version is ECHAM 6 (Stevens et al., 2013). For a detailed list on ECHAM publications we refer to the homepage of the institute (mpimet.mpg.de). Version 5 of the model was used in the 4th Assessment Report of the Intergovernmental Panel on Climate Change (IPCC, 2007), version 6 in the Coupled Model Intercomparison Project CMIP (World Climate Research Programme, 2019a).

Motivation for our work was the usage of ECHAM for long-time paleo-climate simulations in the German national climate modeling initiative "PalMod: From the Last Interglacial to the Anthropocene – Modeling a Complete Glacial Cycle" (www.palmod.de). The aim of this initiative is to perform simulations for a complete glacial cycle, i.e. about 120'000 years, with fully coupled ESMs.



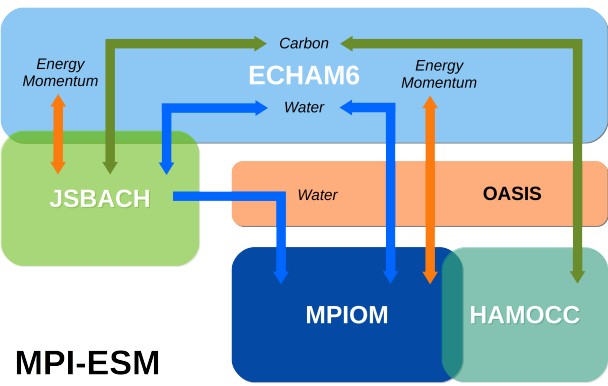

**Figure 1.** Schematic of the structure of the Earth System Model MPI-ESM with atmospheric component ECHAM, terrestrial vegetation model JSBACH, ocean model MPI-OM, marine biogeochemical model HAMOCC, and OASIS coupler.

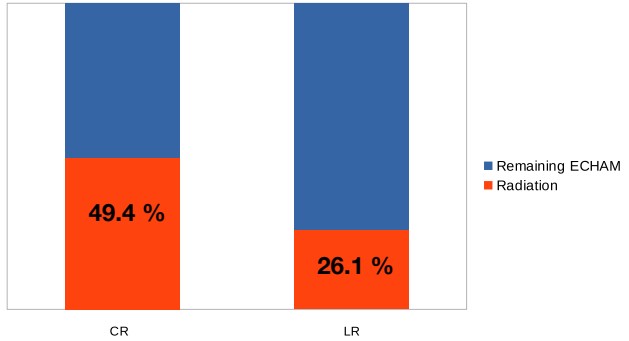

**Figure 2.** Time consumption of the radiation part w.r.t. the whole ECHAM model in coarse (CR) and low resolution (LR) of standard PalMod experiments. The difference occurs since in these configuration the radiation part is called every 2 hours, i.e., only every fourth (in CR) or every eighth time-step (LR).

The feasibility of long-time simulation runs highly depends on the computational performance of the used models. As a consequence, one main focus in the PalMod project is to decrease the runtime of the model components and the coupled ESMs.

In ESMs that use ECHAM, the part of the computational time that is used by the latter is significant. It can be close to 75% in some configurations. Within ECHAM itself, the radiation takes the most important part of the computational time. As a consequence, the radiation part is not called in every time-step in the current ECHAM setting. Still, its part of the overall ECHAM runtime is relevant, see Figure 2.

In the PalMod project, two different strategies to improve the performance of the radiation part are investigated: One is to run the radiation in parallel on different processors, the other one is the conversion to single precision arithmetic we present in





this paper. For both purposes, the radiation code was isolated from the rest of the ECHAM model. This technical procedure is not described here.

The motivation for the idea to improve the computational performance of ECHAM by a conversion to reduced arithmetic
precision was the work of Vana et al. (2017). In this paper, the authors report on the conversion of the Integrated Forecasting System (IFS) model to single precision, observing a runtime reduction of 40% in short-time runs of 12 or 13 months, and a good match of the output with observational data. Here, the terms *single* and *double precision* refer to the IEEE-754 standard for floating point numbers (IEEE Standards Association, 2019), therein also named *binary64* and *binary32*, respectively. In the IEEE standard, also an even more reduced precision, the *half precision (binary16)* format is defined. The IFS model, developed
also by the ECMWF, is comparable to ECHAM in some respect since it also uses a combination of spectral and grid-point-based discretization. A similar performance gain of 40% was reported by Rüdisühli et al. (2014) with the COSMO model that is used by the German and Swiss weather forecast services. The authors also validated the model output by comparing it to observations and the original model version.

Recently, the usage of reduced precision arithmetic has gained interest for a variety of reasons. Besides the effect on the
runtime, also a reduction of energy consumption is mentioned, see, e.g., Fagan et al. (2016), who reported a reduction by about 60%. In the growing field of machine learning, single or even more reduced precision is used to save both computational effort as well as memory, motivated by the usage of Graphical processing Units (GPUs). Dawson and Düben (2017) used reduced precision to evaluate model output uncertainty. For this purpose, the authors developed a software where a variable precision is available, but a positive effect on the model runtime was not their concern.

The process of porting a simulation code to a different precision highly depends on the design of the code and the way how basic principles of software engineering have been followed during the implementation process. These are modularity, use of clear subroutine interfaces, way of data transfer via parameter list or global variables etc. The main problem in legacy codes with a long history (as ECHAM) is that these principles usually were not applied very strictly. This is a general problem in computational science software, not only in climate modeling, see for example Johanson and Hasselbring (2018).

Besides the desired performance gain, a main criterion to assess the result of the conversion to reduced precision is the validation of the results, i.e., their differences to observational data and the output of the original, double precicion version. We carried out experiments on short time scales of 30 years with a 10 years spin-up. It has to be taken into account that after the conversion, a model tuning process (in fully coupled version) might be necessary. This will require a significant amount of work to obtain an ESM that produces a reasonable climate, see, e.g., Mauritsen et al. (2012) for a description of the tuning of
the MPI-ESM.

The structure of the paper is as follows: In the following section, we describe the situation from where our study and conversion started. In Section 3, we give an overview about possible strategies to perform a conversion to single precision, discuss their applicability, and finally the motivation for the direction we took. In Section 4, we describe changes that were necessary at some parts of the code due to certain used constructs or libraries, and in Section 5 the parts of the code that need to
remain in higher precision. In Section 6, we present the obtained results w.r.t. performance gain, output validation and energy consumption. At the end of the paper, we summarize our work and draw some conclusions.



## 2  Used configuration of ECHAM

The current major version of ECHAM, version 6, is described in Stevens et al. (2013). ECHAM is a combination of a spectral and a grid-based finite difference model. It can be used in five resolutions, ranging from the *coarse resolution* (CR) or T31

(i.e., a truncation to 31 wave numbers in the spectral part, corresponding to a horizontal spatial resolution of $96 \cdot 48$ points in longitude and latitude) up to XR or T255. We present results for the CR and LR (*low resolution*, T63, corresponding to $192 \cdot 96$ points) versions. Both use 47 vertical layers and (in our setting) time steps of 30 and 15 min., respectively.

ECHAM6 is written in free format Fortran and conforms to the Fortran 95 standard (Metcalf et al., 2018). It consists of about 240'000 lines of code (including approximately 71'000 lines of the JSBACH vegetation model) and uses a number of

external libraries including LAPACK, BLAS (for linear algebra), MPI (for parallelization), and NetCDF (for in- and output). The radiation part that we converted contains about 30'000 lines of code and uses external libraries as well.

The basis ECHAM version we used is derived from the stand-alone version ECHAM-6.03.04p1. In this basis version, the radiation was modularly separated from the rest of ECHAM. This offers the option to run the radiation and the remaining part of the model on different processors in order to reduce the running time by parallelization, but also maintains the possibility of

running the ECHAM components sequentially. It was shown that the sequential version reproduces bit-identical results with the original code.

All the results presented below are evaluated with the Intel Fortran compiler 18.0 (update 4) (Intel, 2017) on the super-computer HLRE-3 *Mistral*, located at the German Climate Computing Center (DKRZ), Hamburg. All experiments used the so-called *compute* nodes of the machine.

## 3  Strategies for conversion to single precision

In this section we give an overview of possible strategies for the conversion of a simulation code (as the radiation part of ECHAM) to single precision arithmetic. We describe the problems that occurred while applying them to the ECHAM radiation part. At the end, we describe the strategy that finally turned out to be successful. The general target was a version that can be used in both single and double precision with as few changes to the source code as possible. Our goal was to achieve a general

setting of the working precision for all floating point variables at one location in one Fortran module. It has to be taken into account that some parts of the code might require double precision. This fact was already noticed in the report on conversion of the IFS model by Vana et al. (2017).

We will from now on refer to the single precision version as *sp*, and to the double precision version as *dp* version.

### 3.1  Use of a precision switch

One ideal and elegant way to switch easily between different precisions of the variables of a code in Fortran is to use a specification of the `kind` parameter for floating point variables as showed in the following example. For reasons of flexibility, the objective of our work was to have a radiation with such precision switch.





```
    ! define variable with prescribed working precision (wp):
    real(kind = wp) :: x
```

The actual value of `wp` can then be easily switched in the following way:

```
    ! define different working precisions:
    integer, parameter :: sp = 4        ! single precision (4 byte)
    integer, parameter :: dp = 8        ! double precision (8 byte)
    ! set working precision:
integer, parameter :: wp = sp
```

The recommendation mentioned by Metcalf et al. (2018, Section 2.6.2) is to define the different values of the `kind` parameter by using the `selected_real_kind` function. It sets the actually used precision via the definition of the desired number of significant decimals (i.e., mantissa length) and exponent range, depending on the options the machine and compiler offer. This reads as follows:

```
! define precision using significant decimals and exponent range:
    integer :: sign_decimals = 6
    integer :: exp_range = 37
    integer, parameter :: sp = selected_real_kind(sign_decimals, exp_range)
    ...
integer, parameter :: dp = selected_real_kind(...,...)
    ! set working precision:
    integer, parameter :: wp = sp
```

In fact, similar settings can be found in the ECHAM module `rk_mo_kind`, but unfortunately they are not consequently used. Instead, `kind = dp` is used directly in several modules. A somehow dirty workaround, namely assigning the value 4 to `dp`

and declaring an additional precision for actual *dp* where needed, circumvents this problem. Then, compilation was possible after some modifications (concerning MPI and NetCDF libraries and the module `mo_echam_radkernel_cross_messages`). The compiled code was crashing at runtime because of internal bugs triggered by code in the module `rk_mo_srtm_solver` and other parts. These issues were solved later when investigating each code part with the incremental conversion method. The cause of these bugs could not be easily tracked.

**3.2   Source code conversion of most time-consuming subroutines**

As mentioned above, the conversion of the whole ECHAM model code using a simple switch was not successful. Thus, we started to identify the most time-consuming subroutines and functions and converted them by hand. This required the conversion of input and output variables in the beginning and at the end of the respective subroutines and functions. The changes in the code are schematically depicted in Figure 3.



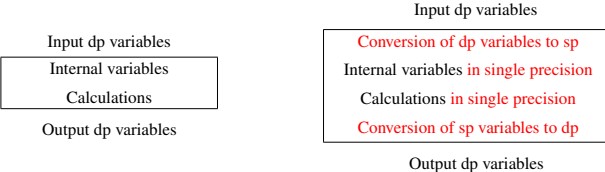

**Figure 3.** Necessary code changes to convert a subroutine/function from the original double precision version (left) to single precision (right) with internal casting, modifications in red.

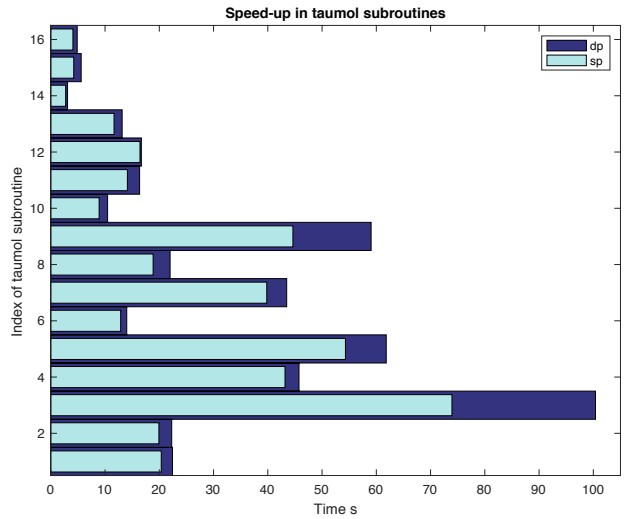

**Figure 4.** Time consumption of single and double precision `taumol` subroutines.

This procedure allowed an effective implementation of *sp* computations of the converted subroutines/functions. We obtained high performance gain in some code parts. But, the casting overhead destroyed the overall performance, especially if there are many variables to be converted.

For example, a time-consuming part of the subroutine `gas_optics_lw` in the module `mo_lrtm_gas_optics` was converted in the above way. The converted part contains calls to subroutines `taumol01` to `taumol16`, which were converted to *sp*. Figure 4 shows the speed-up for these subroutines, which was up to 30%. But the needed casting in the calling subroutine `gas_optics_lw` doubled the overall runtime in *sp*, compared to *dp*.

The results of this evaluation lead to the following conclusion, which is not very surprising: The bigger the converted code block is with respect to the number of input and output variables, the lower the overhead for the casting will be in comparison to the gain in the calculations that are actually performed in *sp*. This was the reason for the decision to convert the whole radiation part of ECHAM, as it contains a relatively small amount of input/output variables.





### 3.3 Incremental conversion of the radiation part

As a result of the not efficient conversion of the most time-consuming subroutines or functions only, we performed a gradual conversion of the whole radiation code. For this purpose, we started from the lowest level of its calling tree, treating each subroutine/function separately. Consider an original subroutine on a lower level,

```
subroutine low(x_dp)
        real(dp) :: x_dp
```

using *dp* variables. We renamed it as `low_dp` and made a copy in *sp*:

```
        subroutine low_sp(x_sp)
        real(sp) :: x_sp
```

We changed the *dp* version such that it just calls its *sp* counterpart, using implicit type conversions before and after the call:

```
        subroutine low_dp(x_dp)
        real(dp) :: x_dp
        real(sp) :: x_sp
        x_sp = x_dp
call low_sp(x_sp)
        x_dp = x_sp
```

Now we repeated the same procedure with each subroutine/function that calls the original `low`, e.g.,

```
        subroutine high(...)
        call low(x_dp)
```

We again renamed it as `high_dp`, generated an *sp* copy `high_sp`, and defined an interface block (Metcalf et al., 2018):

```
        interface low
           module procedure low_sp
           module procedure low_dp
        end interface
```

In both `high_dp` and `high_sp`, we could now call the respective version of the lower level subroutine passing either *sp* or *dp* parameters. The use of the interface simplified this procedure significantly.

If the model output was acceptable, the *dp* version on the lower level, `low_dp`, as well as the interface were redundant, and we deleted them. Finally, the *sp* version could be renamed as `low`.

This procedure was repeated up to the highest level of the calling tree. It required a lot of manual work, but it allowed the 170 examination of each modified part of the code, as well as a validation of the output data of the whole model.





In the ideal case, this would have led directly to a consistent *sp* version. Replacing then the sp by wp in the code for a working precision that could be set to sp or dp in some central module, we would have ended up with a model version that has a precision switch. The next section summarizes the few parts of the code that needed extra treatment.

## 4    Necessary changes in the radiation code

Changing the floating point precision in the radiation code required some modifications that are described in this section. Some of them are related to the use of external libraries, some others to an explicitly used precision-dependent implementation.

### 4.1    Procedure

In the incremental conversion, the precision variables dp and wp that are both used in the radiation code were replaced with sp. Then in the final version, sp was replaced by wp. With this modification, wp became a switch for the radiation precision. As
the original radiation contained several variables lacking explicit declaration of their precision, the respective format specifiers were added throughout.

A Fortran compiler option (namely -real-size 32) can avoid the last step by assigning *sp* to variables without format specifier. However, since the original model used both *sp* and *dp*, this option could lead to an overhead due to conversion in the *dp* part. Although a compilation of the program with custom option was possible for the *sp* Fortran modules, this procedure
led to a more complicated compilation procedure and was therefore discarded. The format specifier D0, denoting *dp*, was also replaced by _wp.

### 4.2    Changes needed for the use of the NetCDF library

In the NetCDF library, the names of subroutines and functions have different suffixes depending on the used precision. They are used in the modules

190        rk_mo_netcdf, rk_mo_cloud_optics, rk_mo_lrtm_netcdf,
        rk_mo_o3clim, rk_mo_read_netcdf77, rk_mo_srtm_netcdf.

In *sp*, they have to be replaced by their respective counterparts to read the NetCDF data with the correct precision. The script shown in Appendix A performs these changes automatically. This solution was necessary because an implementation using an interface was causing crashes for unknown reasons. It is possible that further investigation could lead to a working interface
implementation for these subroutines/functions also at this point of the code.

### 4.3    Changes needed for parallelization with MPI

Several interfaces of the module mo_mpi were adapted to support *sp*. In particular p_send_real, p_recv_real and p_bcast_real were overloaded with *sp* subroutines for the needed array sizes. These modifications did not affect the calls to these interfaces. No conversions are made in this module, so no overhead is generated.





## 4.4 Changes needed due to data transfer to the remaining ECHAM

In ECHAM, data communication between the radiation part and the remaining atmosphere is implemented in the module

```
mo_echam_radkernel_cross_messages
```

through subroutines using both MPI and the coupling library YAXT (DKRZ, 2013). Since it was not possible to have a mixed precision data transfer for both libraries, our solution was to double the affected subroutines to copy and send both *sp* and *dp* data. An additional variable conversion before or after their calls preserves the needed precision. Also in this case, interface blocks were used to operate with the correct precision. The changed subroutines have the following prefixes: `copy_echam_2_kernel`, `copy_kernel_2_echam`, `send_atm2rad` and `send_rad2atm`. These modifications only affect the ECHAM model when used in the parallel scheme. They have a negligible overhead.

## 5 Parts still requiring higher precision

In the *sp* implementation of the radiation code, some parts are still requiring higher precision to run correctly. These parts and the reasons are presented in this section.

### 5.1 Overflow avoidance

When passing from *dp* to *sp* variables, the maximum representable number decreases from $\approx 10^{308}$ to $\approx 10^{38}$. In order to avoid overflow that could lead to crashes, it is necessary to adapt the code to new thresholds. A similar problem could potentially occur for numbers which are too small (smaller than $\approx 10^{-45}$).

As stated in the comments in the original code of `psrad_interface`, the following exponential needs conversion if not used in *dp*:

```
!this is ONLY o.k. as long as wp equals dp, else conversion needed
cisccp_cldemi3d(jl,jk,krow) = 1._wp - exp(-1._wp*cld_tau_lw_vr(jl,jkb,6))
```

One plausible reason for this is that the exponential is too big for the range of *sp*. Even though this line was not executed in the used configuration, we converted the involved quantities to *dp*. Since the variable on the left-hand side of the assignment was transferred within few steps to code parts outside the radiation, no other code inside the radiation had to be converted into *dp*.

Also module `rk_mo_srtm_solver` contained several parts sensitive to the precision. First of all, the following lines containing the hard coded constant 500 could cause overflow as well:

```
exp_minus_tau_over_mu0(:) = inv_expon(MIN(tau_over_mu, 500._wp), kproma)
exp_ktau              (:) =     expon(MIN(k_tau,      500._wp), kproma)
```

Here, `expon` and `inv_expon` calculate the exponential and inverse exponential of a vector (of length `kproma` in this case). The (inverse) exponential of a number close to 500 is too big (small) to be represented in *sp*. In the used configurations, this





line was not executed either. Nevertheless, we replaced this value by a constant depending on the used precision, see the script
in Appendix A.

## 5.2   Numerical stability

Subroutine `srtm_reftra_ec` of module `rk_mo_srtm_solver`, described in Meador and Weaver (1979), showed to be
very sensitive to the precision conversion. In this subroutine, already a conversion to *sp* of just one of most internal variables
separately was causing crashes. We introduced wrapper code for this subroutine to maintain the *dp* version. The time necessary
for this overhead was in the range of 3.5 to 6 % for the complete radiation and between 0.6 and 3% for the complete ECHAM
model.

In subroutine `Set_JulianDay` of the module `rk_mo_time_base`, the use of *sp* for the variable `zd`, defined by

```
zd = FLOOR(365.25_dp*iy)+INT(30.6001_dp*(im+1))  &
     + REAL(ib,dp)+1720996.5_dp+REAL(kd,dp)+zsec
```

caused crashes at the beginning of some simulation years. In this case, the relative difference between the *sp* and the *dp*
representation of the variable `zd` is close to machine precision (in *sp* arithmetic), i.e., the relative difference attains its maximum
value. This indicates that code parts that use this variable afterwards are very sensitive to small changes in input data. The code
block was kept in *dp* by reusing existing typecasts, without adding new ones. Thus, this did not increase the runtime. Rewriting
the code inside the subroutine might improve the stability and avoid the typecasts completely.

## 245  5.3   Quadruple precision

The module `rk_mo_time_base` also contains some parts in quadruple (`REAL(16)`) precision in the subroutine
`Set_JulianCalendar`, e.g.:

```
zb = FLOOR(0.273790700698850764E-04_wp*za-51.1226445443780502865715_wp)
```

Here `wp` was set to `REAL(16)` in the original code. This high precision was needed to prevent roundoff errors because of the
number of digits in the used constants. We did not change the precision in this subroutine. But since we used `wp` as indicator
for the actual working precision, we replaced `wp` by `ap` (advanced precision) to avoid conflicts with the working precision in
this subroutine. We did not need to implement any precision conversion, since all input and output variables are converted from
and to integer numbers inside the subroutine anyway.

## 6   Results

In this section, we present the results obtained with the *sp* version of the radiation part of ECHAM. We show three types of
results, namely a comparison of the model output, the obtained gain in runtime and finally the gain in energy consumption.

The results presented below were obtained with the AMIP experiment (World Climate Research Programme, 2019b) by
using the coarse (CR, T031L47) or low (LR, T063L47) resolutions of ECHAM. The model was configured with the `cdi-pio`





parallel input-output option (Kleberg et al., 2017). We used the following compiler flags (Intel, 2017), which are the default

ones for ECHAM:

- `-O3`: enables aggressive optimization,

- `-fast-transcendentals, -no-prec-sqrt, -no-prec-div`: enable faster but less precise transcendental functions, square roots, and divisions,

- `-fp-model source`: rounds intermediate results to source-defined precision,

- `-xHOST`: generates instructions for the highest instruction set available on the compilation host processor,

- `-diag-disable 15018`: disables diagnostic messages,

- `-assume realloc_lhs`: uses different rules (instead of those of Fortran 2003) to interpret assignments.

## 6.1  Validation of model output

To estimate the output quality of the *sp* version, we compared its results with

- the results of the original, i.e., the *dp* version of the model

- and observational data available from several datasets.

We computed the difference between the outputs of the *sp* and *dp* versions and the differences of both versions to the observational data. We compared the values of

- temperature (at the surface and at 2m height), using the CRU TS4.03 dataset (University of East Anglia Climatic Re-
search Unit et al., 2019),

- precipitation (sum of large scale and convective in ECHAM), using the GPCP data provided by the NOAA/OAR/ESRL PSD, Boulder, Colorado, USA (Adler et al., 2003).

- cloud radiative effect (CRE at the surface and at the top of the atmosphere, the latter split into longwave and shortwave parts), using the CERES EBAF datasets release 4.0 (Loeb and National Center for Atmospheric Research Staff, 2018).

In all results presented below, we use the monthly mean of these variables as basic data. This is motivated by the fact that we are interested in long time simulation runs and climate prediction rather than in short-term scenarios (as for weather prediction). Monthly means are directly available as output from ECHAM.

All computations have been performed with the use of the *Climate Data Operators (CDO)* (Schulzweida, 2019).





| Variable | ECHAM variable | Unit | Time span | $sp - dp$ | $dp - \mathrm{obs}$ | $sp - \mathrm{obs}$ |
|---|---|---|---|---|---|---|
| Surface temperature | 169 | K | 1981 - 2010 | 2.0302 | 2.2862 | 2.2585 |
| Temperature at 2m | 167 | K | 1981 - 2010 | 2.0871 | 2.0585 | 2.0284 |
| Precipitation | 142+143 | $\mathrm{kgm}^{-2}\mathrm{s}^{-1}$ | 1981 - 2010 | $1.9281 \cdot 10^{-5}$ | $2.7200 \cdot 10^{-5}$ | $2.7161 \cdot 10^{-5}$ |
| CRE, surface | 176-185+177-186 | $\mathrm{Wm}^{-2}$ | 2000 - 2010 | 12.3661 | 23.4345 | 23.4753 |
| Shortwave CRE, top of atmosphere | 178-187 | $\mathrm{Wm}^{-2}$ | 2001 - 2010 | 14.3859 | 31.5827 | 31.7220 |
| Longwave CRE, top of atmosphere | 179-188 | $\mathrm{Wm}^{-2}$ | 2001 - 2010 | 9.3010 | 13.2364 | 13.2903 |

**Table 1.** Spatial RMSE of monthly means for *sp* and *dp* versions and difference of both versions in the same metric, for selected model variables, averaged over the respective time spans (obs = observational data).

### 6.1.1 Difference in RMSE between single and double version and observational data

We computed the spatial root mean square error (RMSE) of the monthly means for both *sp* and *dp* versions and the above variables. We applied the same metric for the difference between the outputs of the *sp* and *dp* versions. We computed these values over time intervals where observational data were available in the used datasets. For temperature and precipitation, these were the years 1981-2010, for CRE the years 2000-2010 or 2001-2010. In all cases, we started the computation in the year 1970, having a reasonable time interval as spin-up.

Figure 5 shows the temporal behavior of the RMSE and the differences of *sp* and *dp* version, as they evolve in time. It can be seen that the RMSEs of the *sp* version are in the same magnitude as those of the *dp* version. Also the differences between both versions are of similar or even smaller magnitude. Moreover, all RMSEs and differences do not grow in time. They oscillate but stay in the same order of magnitude for the whole considered time intervals.

Additionally, we averaged these values over the respective time intervals. Table 1 again shows that the RMSEs of the *sp*
version are in the same magnitude as those of the *dp* version. Also the differences between both versions are of similar or smaller magnitude.

Since the RMSE averages spatial differences, we also took a look at the minimum and maximum over all grid-points. This is a check whether the conversion introduced arbitrary biases for the considered variables over the runtime. This method allows only a rough test, since a model producing unreliable data could anyway produce low averages, minima or maxima, even more
if they are averaged annually. This comparison showed a similar magnitude as the differences obtained using two subsequent ECHAM versions. Thus, we do not show corresponding plots here.

Moreover, we compared our obtained differences with the ones between two runs of the ECHAM versions 6.3.02 and 6.3.02p1. The differences between *sp* and *dp* version are in the same magnitude as the differences between these two model versions.





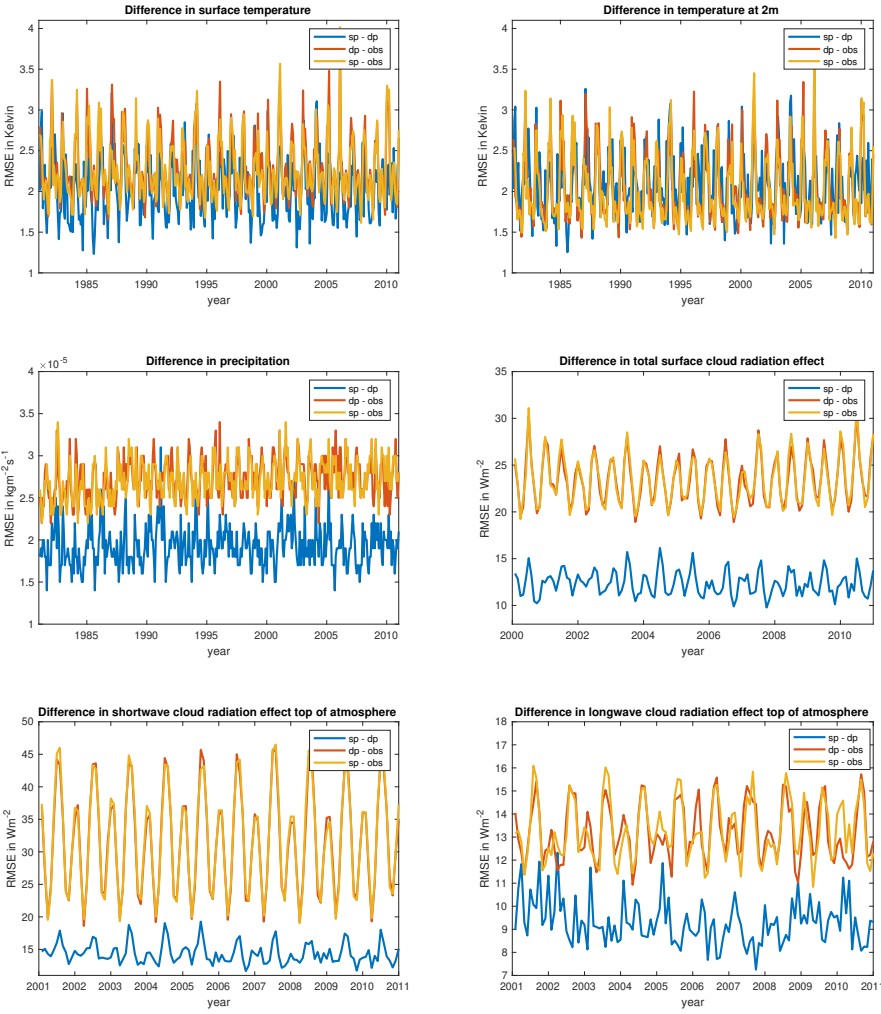

**Figure 5.** Spatial RMSE of monthly means for *sp* and *dp* versions and differences between them in the same metric, for (from top left to bottom right) temperature at the surface and at 2m, precipitation, total CRE at the surface, longwave and shortwave part of CRE at top of the atmosphere.

### 6.1.2 Spatial distribution of differences in the annual means

We also studied the spatial distribution of the differences in the annual means. Again we considered the differences between *sp* and *dp* version and of the output of both versions to the observations. Here we included the signs of the differences and no absolute values or squares. For the given time spans, this results in a function of the form

$$\text{DIFF}(\text{grid-point}) := \frac{1}{\#\text{months in time-span}} \sum_{\text{months in time-span}} \big( y(\text{grid-point}, \text{month}) - z(\text{grid-point}, \text{month}) \big)$$

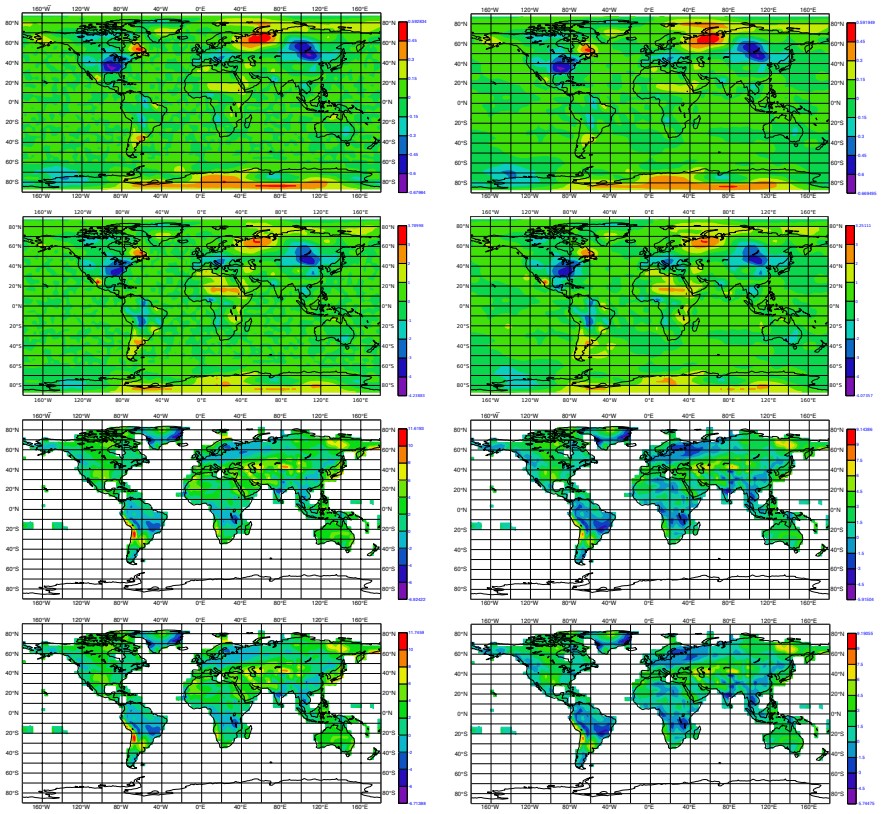

**Figure 6.** Differences in temporal mean over the time interval 1981 to 2010 in temperature at surface (left) and at 2m height (right) in Kelvin. Top: difference between *sp* and *dp* version, second row: values of two-sided t-test with respect to variance of the annual *sp* output, absolute values below 2.05 are not significant at the 95 confidence level, third row: between *dp* version and observational data, bottom: between *sp* version and observational data.

for two variables or datasets $y, z$ of monthly data. This procedure can be used to see if some spatial points or areas are constantly warmer or colder over longer time ranges. It is also a first test of the model output. However, it is clearly not sufficient for validation because errors may cancel out over time. The results are shown in Figures 6 to 8.

Additionally, we performed a statistic analysis of the annual means of the *sp* version. We checked the hypothesis that the 30-years mean (in the interval 1981-2010) of the *sp* version equals the one of the original *dp* version. For this purpose, we applied

a two-sided t-test, using a consistent estimator for the variance of the annual means of the *sp* version. The corresponding values are showed in the respective second rows in Figures 6 to 8. In this test, absolute values below 2.05 are not significant at the 95 confidence level. For all considered variables, it can be seen that only very small spatial regions show higher values.




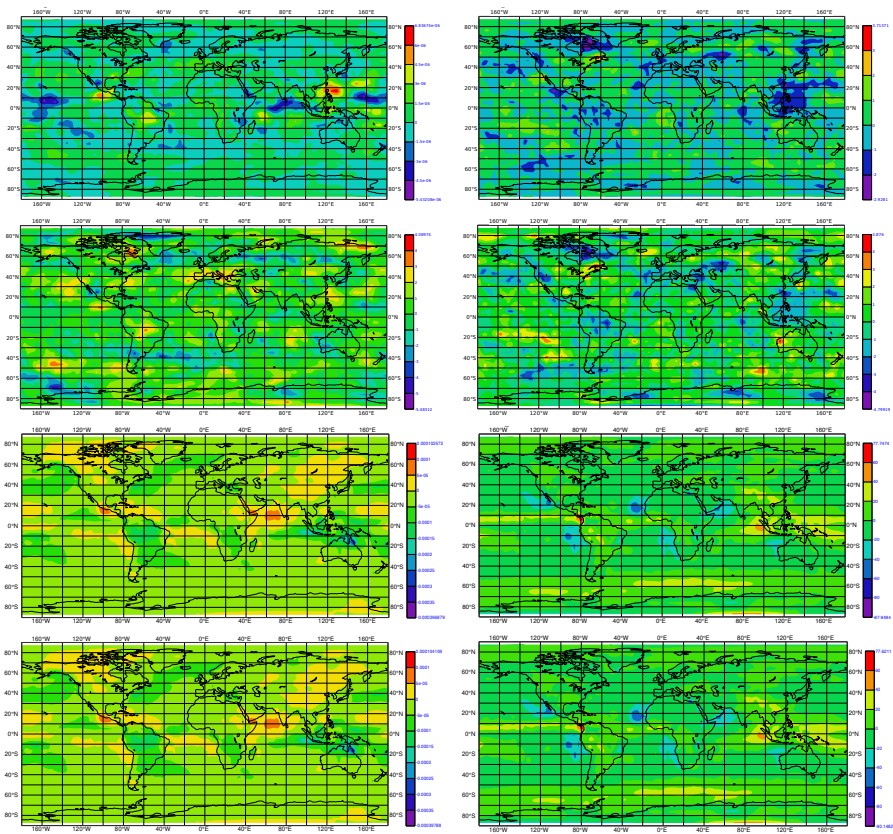

**Figure 7.** As Figure 6, but for precipitation (left) in $\mathrm{kg\,m^{-2}s^{-1}}$ and surface cloud radiation effect in $\mathrm{W\,m^{-2}}$. For the latter difference to observations over 2000-2010.

## 6.2 Speed-up

In this section we present the results of the obtained speed-up when using the modified *sp* radiation code in ECHAM. Since

the model is usually run on parallel hardware, there are several configuration options that might affect its performance and also the speed-up when using the *sp* instead of the *dp* radiation code. We used the Mistral HPC system at DKRZ with 1 to 25 nodes, each of which has two Intel® Xeon® E5-2680v3 12C 2.5GHz ("Haswell") with 12 cores, i.e., using from 24 up to 600 cores. The options we investigated are:

- The number of used nodes.

- The choices *cyclic:block* and *cyclic:cyclic* (in this paper simply referred to as *block* and *cyclic*) offered by the *SLURM* batch system (SchedMD®, 2019) used on Mistral. It controls the distribution of processes across nodes and sockets inside a node.





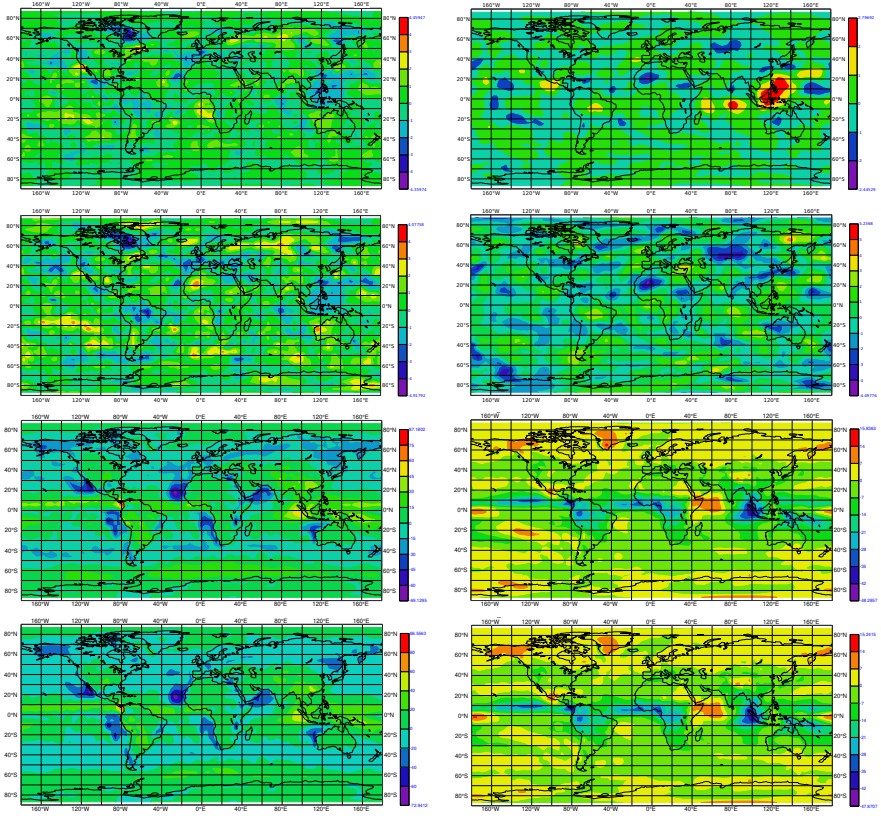

**Figure 8.** As Figure 6, but longwave (left) and shortwave cloud radiation effect at the top of the atmosphere in $\mathrm{Wm}^{-2}$. Difference to observations over 2001-2010 for both.

- Different values of the ECHAM parameter *nproma*, the maximum block length used for vectorization. For a detailed description see Rast (2014, Section 3.8).

We were interested in the best possible speed-up when using the *sp* radiation in ECHAM. We studied the performance gain achieved both for the radiation itself and for the whole ECHAM model for a variety of different settings of the mentioned options, for both CR and (with reduced variety) LR resolutions. Our focus lies on the CR version, since it is the configuration that is used in the long-time paleo runs intended in the PalMod project.

The results presented in this section have been generated with the *Scalable Performance Measurement Infrastructure for*
*Parallel Codes* (Score-P, 2019) and the internal ECHAM timer.

All time measurements are based on one-year runs. The unit we use to present the results is the number of simulated years per day runtime. It can be computed by the time measurements of the one-year runs. For the results for the radiation part only, these are theoretical numbers, since the radiation is not run stand-alone for one year in ECHAM. We include them to give an

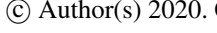



**Table 2.** Relative standard deviations of runtime over 100 runs.

|  | | 24 nodes block | 24 nodes cyclic | 1 node block | 1 node cyclic |
|---|---|---|---|---|---|
| Radiation | dp | 0.0095 | 0.0104 | 0.0081 | 0.0075 |
| | sp | 0.0132 | 0.0122 | 0.0079 | 0.0084 |
| ECHAM | dp | 0.0220 | 0.0179 | 0.0030 | 0.0023 |
| | sp | 0.0158 | 0.0189 | 0.0027 | 0.0020 |

impression what might be possible when more parts of ECHAM or even the whole model would be converted to *sp*. Moreover,
we wanted to see if the speed-up of 40% achieved with IFS model in Vana et al. (2017) could be reached.

To figure out if there are significant deviations in the runtime, we also applied a statistic analysis for 100 one-year runs. They
showed that there are only very small relative deviations from the mean, see Table 2.

At the end of this section, we give some details which parts of the radiation code benefit the most from the conversion to
reduced precision, and which ones not.

### 6.2.1    Dependency of runtime and speed-up on parameter settings

In order to find out the best possible speed-up when using the *sp* radiation code, we first analyzed the dependency of the
runtime on the parameter *nproma*. For the CR resolution, we tested for 1 to 25 cores *nproma* values from 4 to 256 in steps of
4. It can be seen in the two top left pictures in Figure 9 that for 24 nodes there is no big dependency on *nproma* for the original
*dp* version, when looking at radiation only. For the *sp* version, the dependency is slightly bigger, which results in a variety of
the achieved speed-up between 25 to 35%.

When looking at the results for the whole ECHAM model on the two left pictures below in in Figure 9, it can be seen that
the dependency of the speed-up on *nproma* becomes more significant.

Using only one node the performance for the *dp* version decreases with higher *nproma*, whereas the *sp* version does not show
that big dependency. The effect is stronger when looking at the radiation time only than for the whole ECHAM. For very small
values of *nproma*, the *sp* version was even slower than the *dp* version. In particular, the default parameter value (*nproma* = 12)
for the *sp* version resulted in slower execution time than the corresponding *dp* version. An increased value of the parameter
(*nproma* = 48) made *sp* faster, even compared to the fastest *nproma* for *dp* (which was 24).

The difference between the *block* and *cyclic* options are not very significant for all experiments, even though *cyclic* was
slightly faster in some cases. The pictures for *cyclic* (not presented here) look quite similar.
Finally we note that measurements for shorter runs of only one month delivered different optimal values of *nproma*.





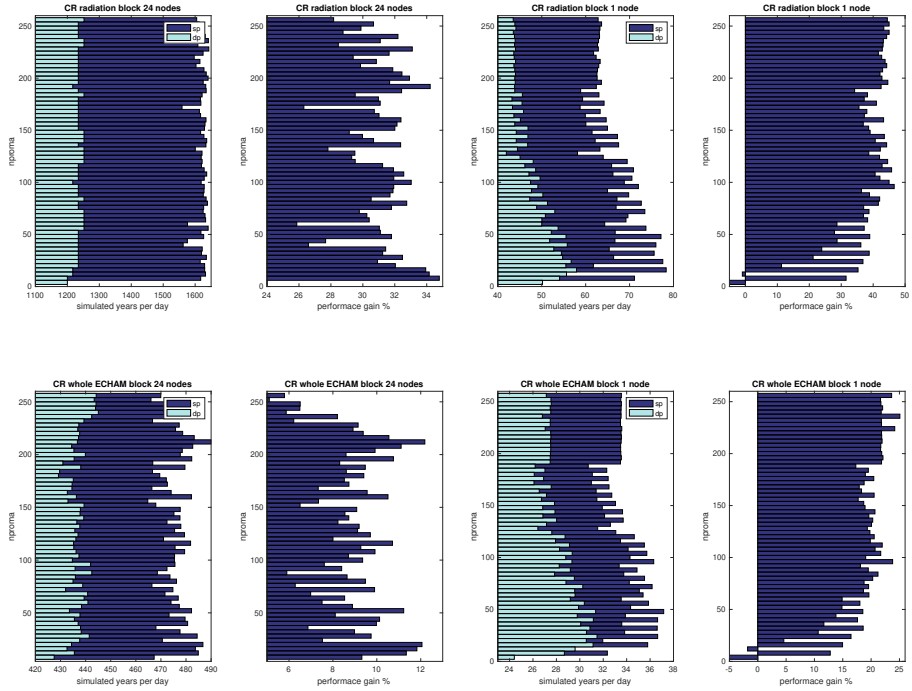

**Figure 9.** Comparison of simulated model years per day cputime for *sp* and *dp* versions in coarse resolution (CR), for radiation part (top) and whole ECHAM (bottom), 1 and 24 nodes and values of *nproma* between 4 and 256, in steps of 4.

### 6.2.2 Best choice of parameter settings for CR configuration

Motivated by the dependency on the parameter *nproma* observed above, we computed the speed-up when using the fastest choice. These runs were performed depending on the number of used nodes (from 1 to 25) in the CR configuration, for both *block* and *cyclic* options. The results are shown in Figure 11. The corresponding best values of *nproma* are given in Tables 3 and 4.

It can be seen that for an optimal combination of number of nodes and *nproma*, the radiation could be accelerated by nearly 40%. On the other hand, a bad choice of processors (here between 16 and 23) results in no performance gain or even a loss.

The speed-up for the whole ECHAM model with *sp* radiation was about 10 to 17%, when choosing an appropriate combination of nodes and *nproma*.

### 6.2.3 Parts of radiation code with highest and lowest speed-up

We identified some subroutines and functions with a very high and some with a very low performance gain by the conversion to sp. They are shown in Tables 5 and 6.



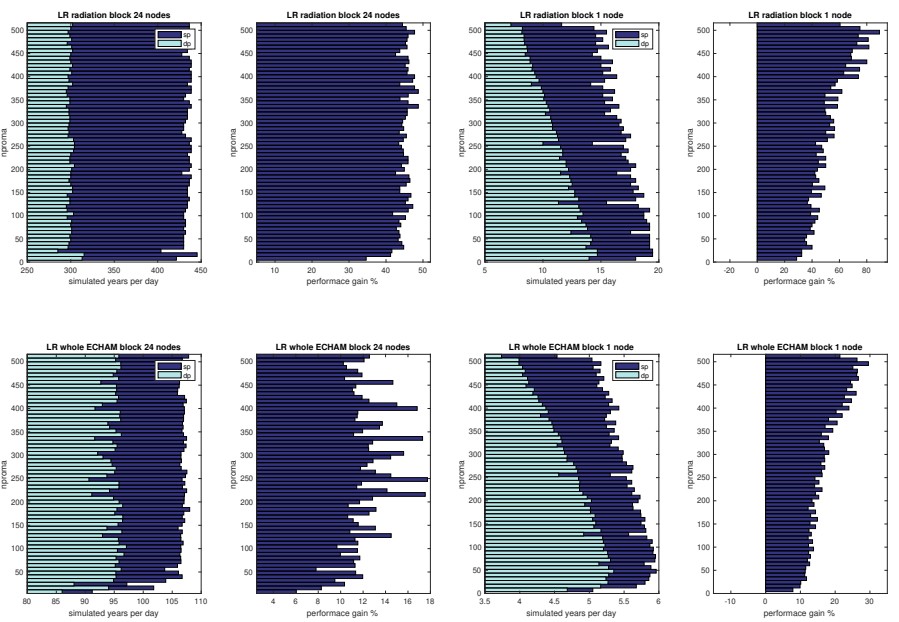

**Figure 10.** As Figure 9, but for low resolution (LR) and values of *nproma* between 8 and 512, in steps of 8.

**Table 3.** Best values of parameter *nproma* for different choice of nodes for radiation part.

| # nodes | 1 | 2 | 3 | 4 | 5 | 6 | 7 | 8 | 9 | 10 | 11 | 12 | 13 | 14 | 15 | 24 | 25 |
|---|---|---|---|---|---|---|---|---|---|---|---|---|---|---|---|---|---|
| *block sp* | 16 | 16 | 16 | 16 | 76 | 16 | 40 | 40 | 36 | 84 | 24 | 16 | 24 | 180 | 36 | 228 | 152 |
| *block dp* | 16 | 16 | 16 | 16 | 20 | 16 | 16 | 56 | 80 | 28 | 156 | 176 | 32 | 28 | 88 | 56 | 32 |
| *cyclic sp* | 48 | 16 | 16 | 24 | 48 | 16 | 16 | 96 | 52 | 188 | 28 | 16 | 124 | 164 | 16 | 252 | 212 |
| *cyclic dp* | 16 | 24 | 16 | 16 | 20 | 16 | 16 | 152 | 172 | 136 | 148 | 32 | 40 | 124 | 16 | 20 | 52 |

A cause that no even higher speed-up was achieved is that several time-consuming parts (as in `rk_mo_random_numbers`) use expensive calculation with integer numbers, taking over 30% of the total ECHAM time consumption in some cases. Therefore, these parts are not affected by the *sp* conversion.

## 6.3 Energy Consumption

We also carried out energy consumption measurements. We used the *IPMI (Intelligent Platform Management Interface)* of the *SLURM* workload manager *ADD* (SchedMD®, 2019). It is enabled with the experiment option

```
#SBATCH --monitoring=power=5
```

Here we used one node with the corresponding fast configuration for *nproma* and the option *cyclic*. Simulations were repeated 10 times with a simulation interval of one year.





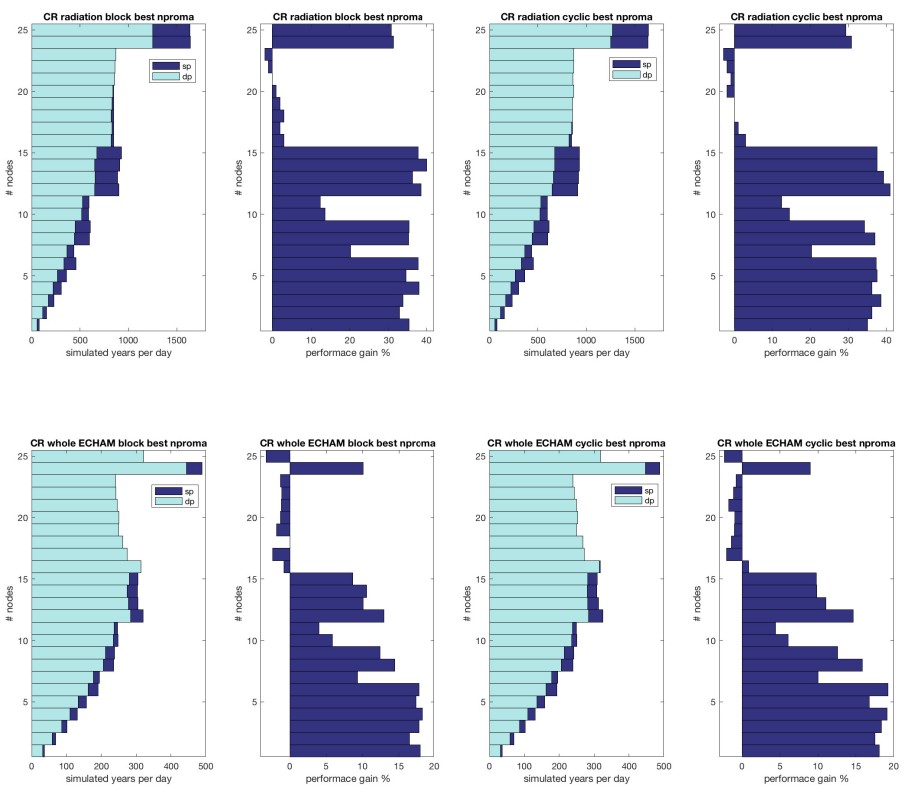

**Figure 11.** As Figure 9, but for 1 to 25 nodes using the respective best value of *nproma*. Cooresponding optimal values can be found in Tables 3 and 4.

**Table 4.** Best values of parameter *nproma* for different choice of nodes for whole ECHAM.

| # nodes | 1 | 2 | 3 | 4 | 5 | 6 | 7 | 8 | 9 | 10 | 11 | 12 | 13 | 14 | 15 | 24 | 25 |
|---|---|---|---|---|---|---|---|---|---|---|---|---|---|---|---|---|---|
| *block sp* | 48 | 48 | 32 | 116 | 72 | 120 | 136 | 44 | 36 | 152 | 120 | 56 | 24 | 236 | 52 | 212 | 48 |
| *block dp* | 24 | 24 | 32 | 68 | 148 | 16 | 88 | 220 | 84 | 232 | 84 | 124 | 252 | 36 | 228 | 240 | 228 |
| *cyclic sp* | 48 | 48 | 32 | 124 | 208 | 72 | 192 | 100 | 220 | 152 | 184 | 100 | 92 | 136 | 68 | 208 | 32 |
| *cyclic dp* | 24 | 24 | 16 | 24 | 200 | 40 | 40 | 160 | 72 | 128 | 144 | 132 | 28 | 60 | 84 | 52 | 204 |

As Table 7 shows, the obtained energy reduction was 13% and 17% in blade and cpu power consumption, respectively. We consider these measurements only as a rough estimate. A deeper investigation of energy saving was not the focus of our work.



**Table 5.** Parts of radiation code with highest speed-up by conversion to sp.

| Module name | Subroutine/Function name | Time *dp* (s) nproma=24 | Time *sp* (s) nproma=48 | Speed-up (%) |
|---|---|---|---|---|
| rk_mo_srtm_solver | delta_scale_2d | 960.27 | 413.35 | 56.95 |
| rk_mo_echam_convect_tables | lookup_ua_spline | 17.42 | 7.67 | 55.97 |
| rk_mo_rrtm_coeffs | lrtm_coeffs | 78.59 | 37.06 | 52.84 |
| rk_mo_lrtm_solver | lrtm_solver | 9815.12 | 4663.04 | 52.09 |
| rk_mo_srtm_solver | srtm_solver_tr | 5005.70 | 2425.09 | 51.55 |
| rk_mo_radiation | gas_profile | 27.69 | 13.57 | 50.99 |
| rk_mo_rad_fastmath | tautrans | 3455.26 | 1790.45 | 50.53 |
| rk_mo_rad_fastmath | transmit | 2837.84 | 1503.41 | 47.02 |
| rk_mo_o3clim | o3clim | 87.10 | 47.32 | 45.67 |
| rk_mo_aero_kinne | set_aop_kinne | 233.65 | 127.88 | 45.27 |

**Table 6.** Parts of radiation code with lowest speed-up by conversion to sp.

| Module name | Subroutine/Function name | Time *dp* (s) nproma=24 | Time *sp* (s) nproma=48 | Speed-up (%) |
|---|---|---|---|---|
| rk_mo_lrtm_gas_optics | gas_optics_lw | 6517.89 | 5647.92 | 13.15 |
| rk_mo_lrtm_solver | find_secdiff | 232.42 | 209.15 | 10.01 |
| rk_mo_random_numbers | m | $1.94 \cdot 10^4$ | $1.83 \cdot 10^4$ | 5.67 |
| rk_mo_random_numbers | kissvec | $8.22 \cdot 10^4$ | $7.79 \cdot 10^4$ | 5.23 |
| rk_mo_lrtm_driver | planckfunction | 2169.69 | 2070.20 | 4.59 |
| rk_mo_srtm_gas_optics | gpt_taumol | 4117.74 | 3931.92 | 4.51 |
| rk_mo_random_numbers | low_byte | $1.36 \cdot 10^4$ | $1.33 \cdot 10^4$ | 2.20 |

## 7 Conclusions

385  We have successfully converted the radiation part of ECHAM to single precision arithmetic. All relevant part of the code can now be switched from double to single precision by setting a Fortran `kind` parameter named `wp` either to `dp` or `sp`. There is one exception where a renaming of subroutines has to be performed. This can be easily done using a (provided) shell script before the compilation of the code.

We described our incremental conversion process in detail and compared it to other, in this case unsuccessful methods.

390  We tested the output for the single precision version and found a good agreement with measurement data. The deviations over decadal runs are comparable to the ones of the double precision versions. The difference between the two version lie in the same range.





**Table 7.** Energy reduction when using *sp* radiation in ECHAM.

| Energy Consumption | ECHAM with *dp* radiation (J) | ECHAM with *sp* radiation (J) | Saved energy (%) |
|:---:|:---:|:---:|:---:|
| Blade power | 803368 | 698095 | 13.1 |
| CPU power | 545762 | 452408 | 17.1 |

We achieved an improvement in runtime in coarse and low and resolution of up to 40% for the radiation itself, and about 10 to 17% for the whole ECHAM. In this respect, we could support results obtained for the IFS model by Vana et al. (2017), where the whole model was converted. We also measured an energy saving of about 13 to 17%.

Moreover, we investigated the parts of the code that are sensitive to reduced precision, and those parts which showed comparably high and low runtime reduction.

The information we provide may guide other people to convert even more parts of ECHAM to single precision. Moreover, they may also motivate to consider reduced precision arithmetic in other simulation codes.

As a next step, the converted model part will be used in coupled ESM simulation runs over longer time horizons.

*Code and data availability.* The code is available in the DKRZ git repository under the URL https://gitlab.dkrz.de/PalMod/echam6-PalMod/ ~/network/mixed_precision_new2 upon request. The conversion scripts (see Appendix A) and the output data for the single and double precision runs that were used to generate the output plots are available as NetCDF files under https://doi.org/10.5281/zenodo.3560536.

## Appendix A:  Conversion Script

The following shell script converts the source code of ECHAM from double to single precision. It renames subroutines and function from the NetCDF library, changes a constant in the code to avoid overflow (in a part that was not executed in the used setting), and sets the constant `wp` that is used as Fortran `kind` attribute to the current working precision, either `dp` or `sp`. A script that reverts the changes is analogous. After the use of one of the two scripts, the model has to be re-compiled. These scripts cannot be used on the standard ECHAM version, but on the one mentioned in the code availability section.

```
#!/bin/bash
# script rad_dp_to_sp.sh
# To be executed from the root folder of ECHAM before compilation
for i in ./src/rad_src/rk_mo_netcdf.f90
        ./src/rad_src/rk_mo_srtm_netcdf.f90
        ./src/rad_src/rk_mo_read_netcdf77.f90
        ./src/rad_src/rk_mo_o3clim.f90
        ./src/rad_src/rk_mo_cloud_optics.f90;
```





```
      do
         sed -i 's/_double/_real/g' $i
sed -i 's/_DOUBLE/_REAL/g' $i
      done
      sed -i 's/numthresh = 500._wp/numthresh = 75._wp /g'
            ./src/rad_src/rk_mo_srtm_solver.f90
      sed -i 's/INTEGER, PARAMETER :: wp = dp/INTEGER, PARAMETER :: wp = sp/g'
./src/rad_src/rk_mo_kind.f90
      echo "ECHAM radiation code converted to single precision."
```

*Author contributions.* The first author performed all experiments, generated a part of the plots and tables and wrote parts of the manuscript. The second author generated the other part of the plots and tables and wrote main parts of the manuscript.

*Competing interests.* The authors declare that no competing interests are present.

*Acknowledgements.* This work was supported by the German Federal Ministry of Education and Research (BMBF) as a Research for Sustainability initiative (FONA) through the project PalMod (FKZ: 01LP1515B), Working Group 4, Work Package 4.4. The authors wish to thank Mohammad Reza Heidari, Hendryk Bockelmann and Jörg Behrens from the German Climate Computing Center DKRZ in Hamburg, Uwe Mikolajewicz and Sebastian Rast from the Max-Planck Institute for Meteorology in Hamburg, Peter Düben from the ECMWF in Reading, Robert Pincus from Colorado University, Zhaoyang Song from the Helmholtz Centre for Ocean Research Kiel GEOMAR and
Gerrit Lohmann from AWI Bremerhaven and Bremen University for their valuable help and suggestions.





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
