# Peer review of "Single precision arithmetic in ECHAM radiation reduces runtime and energy consumption"

_Geoscientific Model Development, 2020_

## Referee Comment (RC1) · Anonymous Referee #1 · 17 Feb 2020

Since the memory access became the most important bottleneck to a numerical simulation efficiency the use of reduced precision becomes an attractive way to gain a speed-up.

Authors very clearly and in good level of detail describe their attempt to use this optimization method for the ECHAM model radiation scheme in this paper. The text is well written, easy to follow and the argumentation behind the subsequent steps is very clear. Finally the achieved speed-up is consistent to what has been already published while the results are impressively comparable across the two compared numerical precisions. Certainly, this is the description of another success story and documentation

of a hard work when getting there.

My main concerns are twofold: First, by their incremental method authors identified routine(s) being problematic with respect to the reduced precision (like for example srtm_reftra_ec). Those were then kept entirely in the double precision. Authors apparently didn't dare to introduce any alternative method mathematically quasi-equivalent but possessing higher robustness with respect to the numerical precision. Similarly they did no attempt to modify the exponentials (inv_expon, expon) evidently performing computation outside the representativeness of 32bits arithmetic. Although those are probably minor issues a curious reader would like to see some discussion about possible entirely singe precision alternative and its eventual cost with respect to the proposed solution to keep it in double precision. Having that discussed the paper would substantially improve its scientific content for a reader outside the ECHAM community. The presented work looks rather like a mechanical task suitable for graduate students. This however is also being of a scientific value as it gives some guidelines for other applications. Still the presented method leaves some impression that when the ECHAM model is partitioned differently into different number of subroutines the resulting code ready for single precision execution might be different.

Second, having some experience from converting few double precision radiation schemes to single precision, it is bit hard to digest the author's claim about being it relatively a straightforward task. Our experience was entirely different. This implies some question-marks about the evaluation method used. Did authors ever evaluated the single precision radiation code separately? It would be for example very interesting to see some single column model simulations targeted to radiation scheme comparing outputs from double and single precision radiations schemes.

Despite those two above comments I still consider the paper to be bringing an interesting material worth for being published.

---

## Referee Comment (RC2) · Anonymous Referee #2 · 3 Mar 2020

The authors comprehensively describe the conversion of the radiation code of ECHAM from double-precision to single-precision (strictly speaking, a model version that allows both double and single-precision). They demonstrate that the model performs well both when compared with the double-precision version and a number of observational datasets. They also assess the computational speed-up and the energy savings.

The paper builds on earlier work on the ECMWF model and COSMO and so is not entirely novel. Nevertheless, there are considerably more technical details than earlier papers and so I recommend it for publication as a useful guide for those developing reduced-precision versions of their own models. This recommendation is subject to

the minor revisions below.

**1  Minor revisions**

- Section 3.3: After reading this section several times, I still don't understand the conversion process. For example:

    - line 167: "If the model output was acceptable" — how do you define "acceptable"? And what do you do if the model output is not acceptable? Do you then delete the sp version?
    - line 167: "low_dp, as well as the interface were redundant" — even if the model output when using low_sp is acceptable, you still need an interface "on top of" low_sp that allows double-precision arguments. This is the purpose of low_dp. If you delete low_dp, how does high_dp call low_sp?

    It could be that the authors won't understand my questions because my thinking is so wrong. In any case, I didn't understand it and I recommend that the authors rewrite this section so the procedure is clearer, perhaps including some diagrams.

- line 182: "namely -real-size 32": this depends on the compiler. I'm not even sure GNU Fortran has an option to set the default REAL precision to 4 bytes, as this is already the language standard (as far as I'm aware). I don't think you need this paragraph at all — you can simply say that you added type declarations to all REAL variables and literals so that the type was always explicit. This is good programming practice anyway.

- lines 297 - 300: I didn't understand this paragraph. For example, "we also took a look at the minimum and maximum over all grid points" — minimum and

maximum error, or minimum and maximum field values? I'm assuming the latter. If so, why does a difference in minimum and maximum indicate a bias? If single-precision has both a larger maximum and smaller minimum than double-precision, the mean could still be zero (meaning zero bias). I recommend either rewriting this paragraph or just deleting it.

- Section 6.1.2: The equation only computes an annual mean if #months in time span $= 12$. In fact the period is 30 years so I think you mean "temporal mean" not "annual mean". The caption of Figure 6 even uses that name.

- Figures 6, 7, 8: This could simply be GMD's formatting but I can't read the colorbars in these figures. If the authors deliberately chose this size for the Figures, please enlarge the colorbars.

- line 359: so all of the shown results are for "block"? Please clarify this.

- The terms "performance gain", "runtime reduction", "speed-up" and "acceleration" are used interchangeably throughout the manuscript (mainly the first three) but it's not clear what they actually mean. If $x$ and $y$ are the wall-clock times for single and double, respectively, is the performance gain (or whatever) $1 - x/y$ or $y/x$? I recommend using the phrase "runtime reduction", meaning $1 - x/y$, as much as possible, as this is what others like Váňa et al. use. "Speed-up" sounds like $y/x$ to me i.e. if $x = 5$ seconds and $y = 10$ seconds, the speed-up is 2 because single is twice as fast.

**2 Typographical comments**

- line 56, typo: double-precicion $\rightarrow$ double-precision

- line 142, language: not efficient → inefficient

- line 351, typo: below in in Figure → below in Figure

- Figures 9, 10, 11, typo: "performace gain" → "performance gain"

- line 373, language: A cause that no higher speed-up was achieved is that → We could not achieve a significant speed-up in some cases because

- line 391, typo: two version → two versions

- line 393, typo: low and resolution → low resolution

---

## Author Comment (AC1) · 17 Apr 2020

We would like to thank the reviewer for the work and his/her comments. We have the following remarks corresponding to the points stated in section 1 of the interactive comments:

Reviewer's comment: Section 3.3: After reading this section several times, I still don't understand the conversion process. For example:

– line 167: "If the model output was acceptable" — how do you define "acceptable"? And what do you do if the model output is not acceptable? Do you then delete the sp

version?

Answer: Here we changed the text to the following: We then tested if the model with the sp version of the subroutine/function compiles, produces no runtime errors, and if its difference to the dp version was in an "acceptable" range. Of course, the latter is a soft criterion, since a bit-identical result cannot be expected. Our criteria are explained below in Subsection 6.1. If the sp output was not acceptable in this sense, we marked the corresponding code part as to be treated separately, as described below in Sections 4 and 5.

Reviewer's comment: – line 167: "low_dp, as well as the interface were redundant" — even if the model output when using low_sp is acceptable, you still need an interface "on top of" low_sp that allows double-precision arguments. This is the purpose of low_dp. If you delete low_dp, how does high_dp call low_sp? It could be that the authors won't understand my questions because my thinking is so wrong. In any case, I didn't understand it and I recommend that the authors rewrite this section so the procedure is clearer, perhaps including some diagrams.

Answer: This was formulated in a misleading way in our first version. The two versions low_sp and low_dp can be deleted only at the end, when a new "wp" version of "low" is introduced, wp being a variable that could be set to "sp" or "dp". Then, also the interface becomes redundant.

Reviewer's comment: line 182: "namely -real-size 32": this depends on the compiler. I'm not even sure GNU Fortran has an option to set the default REAL precision to 4 bytes, as this is already the language standard (as far as I'm aware). I don't think you need this paragraph at all — you can simply say that you added type declarations to all REAL variables and literals so that the type was always explicit. This is good programming practice anyway.

Answer: Paragraph was omitted.

Reviewer's comment: lines 297 - 300: I didn't understand this paragraph. For example, "we also took a look at the minimum and maximum over all grid points" — minimum and maximum error, or minimum and maximum field values? I'm assuming the latter. If so, why does a difference in minimum and maximum indicate a bias? If single-precision has both a larger maximum and smaller minimum than doubleprecision, the mean could still be zero (meaning zero bias). I recommend either rewriting this paragraph or just deleting it.

Answer: We removed the paragraph.

Reviewer's comment: Section 6.1.2: The equation only computes an annual mean if #months in time span = 12. In fact the period is 30 years so I think you mean "temporal mean" not "annual mean". The caption of Figure 6 even uses that name.

Answer: We added an explaining sentence after the formula.

Reviewer's comment: Figures 6, 7, 8: This could simply be GMD's formatting but I can't read the colorbars in these figures. If the authors deliberately chose this size for the Figures, please enlarge the colorbars.

Answer: We enlarged the whole figures to make all detailed better visible.

Reviewer's comment: line 359: so all of the shown results are for "block"? Please clarify this.

Answer: We added this at (new) line 360 (old line 359) and also before in (new) line 328.

Reviewer's comment: The terms "performance gain", "runtime reduction", "speed-up" and "acceleration" are used interchangeably throughout the manuscript (mainly the first three) but it's not clear what they actually mean. If x and y are the wall-clock times for single and double, respectively, is the performance gain (or whatever) $1 - x/y$ or $y/x$? I recommend using the phrase "runtime reduction", meaning $1 - x/y$, as much as possible, as this is what others like Vána et al. use. "Speed-up" sounds like ĚĞ $y/x$ to

[Figure]

**GMDD**

me i.e. if x = 5 seconds and y = 10 seconds, the speed-up is 2 because single is twice as fast.

Answer: We followed the suggestion to use "(relative) runtime reduction" all the time and defined the term as 1-x/y = (dp-sp)/dp, at the beginning of subsection 6.2.

Answer to reviewer's section 2: All mentioned typos were corrected.

─────────────────────────────

---

## Author Comment (AC2) · 17 Apr 2020

We would like to thank the reviewer for the work and his/her comments. We have the following remarks corresponding to the two main points he/she stated in the interactive comments:

We understood the first point the reviewer made in the following way: We did not investigate how to replace those subroutines or code parts that still require double or even quadruple precision (mentioned in Section 5) by alternative code. This is completely correct; it was due to the scope of the third-party-funded project this investigation was conducted in. For the 2nd version of the manuscript, we now included some remarks

that point this out, and mentioned it as an option for future work. This text was inserted at the beginning of Section 5 and in the conclusions.

Concerning the second point the reviewer made, we understood the following: We only validated the results of the whole model after changing the radiation to single precision. We never tested the two radiation versions (sp and dp) alone or in some kind of idealized configuration (e.g., in single atmospheric column configurations). This is also true, and we also agree that this would be a validation that would be much more exact. The reason or answer is similar to above: Also this was due to the scope of the third-party-funded project this investigation was conducted in, which was very much result-oriented regarding simulation runs for coupled models. We added remarks on this point directly in front of Subsection 6.1.1.

―――――――――――――――